# "I'm in a very good frame of mind": a qualitative exploration of the experience of standing frame use in people with progressive multiple sclerosis

Rachel Dennett [1], Wendy Hendrie,[2] Louise Jarrett,[3] Siobhan Creanor,[4] Andrew Barton,[5] Annie Hawton,[6] Jennifer A Freeman[1]

¹School of Health Professions, Faculty of Health, University of Plymouth, Plymouth, UK
²Physiotherapy, MS Therapy Centre, Norwich, UK
³Mardon Neurorehabilitation Centre, Royal Devon and Exeter NHS Foundation Trust, Exeter, UK
⁴Medical Statistics Group, Faculty of Health, University of Plymouth, Plymouth, UK
⁵Research Design Service, Faculty of Health, University of Plymouth, Plymouth, UK
⁶Health Economics Group, University of Exeter Medical School, University of Exeter, Exeter, UK

**Correspondence to**
Dr Jennifer A Freeman;
jenny.freeman@plymouth.ac.uk

## ABSTRACT

**Objectives** The study aim was to explore the experiences of people with progressive multiple sclerosis (MS) and their standing assistants during their participation in Standing Up in Multiple Sclerosis, a randomised controlled trial (RCT) of a home-based, self-managed standing frame programme.

**Design** A qualitative approach, using audio diary methodology was used to collect data contemporaneously. Diary data were transcribed verbatim and analysed using thematic analysis.

**Setting** Participants were recruited from eight healthcare organisations in two regions of the UK. The intervention was home-based.

**Participants** As part of the RCT, 140 participants were randomly allocated to either usual care or usual care plus a standing frame programme. Using a sampling matrix 12 people with progressive MS (6 female, aged 35–71 years, Expanded Disability Status Scale 6.5–8.0) and 8 standing assistants (4 female) kept audio diaries of their experiences.

**Intervention** The standing frame programme involved two face-to-face home-based physiotherapy sessions to set up the standing frame programme, supplemented by educational material designed to optimise self-efficacy. Participants were encouraged to stand for at least 30 min, three times a week for the 36-week study period.

**Results** Four main themes were identified: "Feeling like the old me"; 'Noticing a difference'; "I want to do it right" and "You have a good day, you have a bad day".

**Conclusions** Supported standing helped people with progressive MS feel more like their old selves and provided a sense of normality and enjoyment. People noticed improvements in physical and psychological symptoms, which were often associated with increased participation in activities they valued. Provision of support from a physiotherapist and recognition of the variable nature of the condition were highlighted as factors to consider when establishing a standing programme.

**Trial registration number** ISRCTN69614598.

## INTRODUCTION

Multiple sclerosis (MS) is a progressive, neurological condition where inflammation and neurodegeneration in the central nervous system can result in a wide variety of symptoms.[1] Approximately 1 000 000[2] people worldwide live with progressive MS, where increasing disability can negatively impact on function and quality of life. With symptom onset commonly in early adulthood and survival rates improving, years lived with disability are increasing.[3] Consequently, people with MS are likely to have an increasing requirement for rehabilitation over the course of their lives to help manage symptoms and maximise independence. Higher levels of disability can make it difficult for people to engage in sufficient physical activity to achieve recognised health benefits,[4–7] with many people with progressive MS spending most of the day sitting down.[4 5 8] This prolonged immobility places them at risk of developing preventable secondary complications, which may include muscle wasting, spasms, constipation and depression.[5]

Effective long-term, physical activity strategies, which can be self-managed and implemented relatively easily and cheaply within people's homes, are needed. The use of a standing frame is one option that enables people with severe MS to increase their physical activity through regular supported standing. Standing frames are devices which

allow people who have limited or no ability to stand upright independently, to do so safely, with good postural alignment and for extended periods of time. We have provided robust evidence, from a definitive, multicentre, randomised controlled trial (RCT) with cost-effective analysis, that a home-based, self-managed standing frame programme, set up by a physiotherapist and supported using behavioural interventions can significantly increase motor function in people with severe progressive MS, is feasible to implement and appears cost-effective.[9] Of importance, at 6 months follow-up, the majority of people allocated to the standing frame group (66%) were continuing to use the frame regularly.

### Strengths and limitations of this study

► Audio diary methodology enabled data to be collected contemporaneously and longitudinally.
► Data include the perspectives of both the person with multiple sclerosis (MS) and their assistant.
► This qualitative data complement and support findings from our randomised controlled trial that demonstrated standing frame use to be beneficial for people with severe progressive MS.
► Some practical difficulties were encountered when using the audio recorders.

Understanding the subjective perspective of using supported standing devices is also important. This has been explored in a small number of studies. Two cross-sectional surveys of frame users with spinal cord injury included questions about the perceived benefits of standing,[10 11] both finding that the vast majority of participants (76% and 87%, respectively) reported improved well-being and quality of life, and that standing had a positive impact on self-esteem and self-image.[10] This improvement in well-being increased to 88% for respondents who stood more than once a day.[12] Similar findings have been reported in surveys of mixed neurological populations, including people with MS.[13 14]

Nordström et al[15] interviewed 15 people (7 of whom had a progressive neurological condition including MS), who had used a variety of standing devices for between 1 and 10 years. The authors described how the upright standing position alters the person's sense of self, augments their availability to the outside world, strengthens social interplay and changes a person's motivation and expectations over time. They concluded that standing unites body to self and emphasised that therapists should understand both the subjective and physiological impacts. Similar conclusions were drawn by Hendrie et al,[16] who used a mixed-methods approach involving nine single-case studies of people with MS who participated in a home-based standing-frame programme. In this study, in-depth interviews were undertaken on three occasions: at baseline (before standing began) and again at 36 and 48 weeks postbaseline. Respondents' stories revealed how regular frame standing enabled them to reconnect with their body, regain skills, re-engage with relationship roles and develop a sense of optimism for the future.

To our knowledge, no previous study has explored the contemporaneous experience of using a standing frame from the initial stand onwards, either from the perspective of the participant or the individual assisting with frame use (referred to here as the 'standing assistant' and typically their spouse). There are a number of important factors to consider in order to optimise any impact of the intervention and subsequently increase the likelihood that this self-managed activity is sustained over the long term. These include: understanding the immediate experiences of using the frame; changes experienced over time; factors which impact on its everyday use and how

standing frames can be integrated into people's everyday lives. We embedded a qualitative component within our RCT,[9] which explored participants' subjective experience of self-managing this standing frame programme, over the 36-week trial period, from the moment the person with MS first stood in the frame. Capturing the personal experience and impact of using a standing frame within daily life was considered important to complement the objective data gathered, and to provide a richer understanding of both the benefits and drawbacks of this intervention.

## MATERIALS AND METHODS
Our multicentre RCT involved 140 people with progressive MS. Abilities ranged from being able to walk 20 m with bilateral assistance to full-time wheelchair users (graded 6.5–8.0 on the Expanded Disability Status Scale (EDSS)). Here, we focus on reporting the embedded qualitative component; the RCT methodology and results have been previously described.[9 17]

The qualitative component reported here is described in line with the standards for reporting qualitative research.[18]

## PATIENT AND PUBLIC INVOLVEMENT
People with MS and their family members were actively involved in development of the research questions, study design, trial management and steering groups, writing study materials and dissemination activities.

### Research approach and methodology
We conducted this qualitative study from a critical realist perspective[19] and explored people's experiences of a particular phenomenon (frame standing) in a particular context (the home environment). We chose the audio diary method to capture data about participants' day-to-day experiences because of its potential ability to reveal people's ongoing, everyday experiences longitudinally by offering 'multi-occasional windows' for data collection.[20 21] We considered this important since people's views and experiences of undertaking an activity alter over time; the mastery of a skill such as standing within the frame is an ongoing process, and MS symptoms fluctuate. The hope, therefore, was that the diaries would enable the immediacy of the moment to be reflected, rather than participants recollecting these feelings later, at a single point in time distant to the event, as would have been the case using interview or focus group methods. Other advantages of the unstructured nature of this approach is that it allows people flexibility over when, where and what to record, and being able to erase files they do not wish to share.[22] Furthermore, the participant is not required to write down their thoughts, which may be problematic for people with progressive MS where upper limb dexterity issues are commonplace.[23] In addition, such a method reduces the bias that may be introduced due to an interviewer's questioning.[21] Exploring the experiences of

both people with MS and standing assistants was considered important due to the invaluable role assistants play in the lives of many people with MS. For many people with significant disability, participation in this standing activity would only be possible with help from an assistant. Understanding both perspectives of the frame use therefore was recognised as crucial for successful integration of a standing programme into everyday life.

### The standing intervention
Briefly, the standing frame programme involved provision of an Oswestry standing frame (Theo Davies & Sons, Wrexham, UK), at home to 71 participants who were allocated to the standing group. A physiotherapist visited the person in their home on two occasions, 1 week apart, to teach the person with MS and their standing assistant how to use the frame. This was complemented by the provision of educational materials, via written and video mediums, designed to optimise self-efficacy (for details, see https://www.plymouth.ac.uk/research/sums). These face-to-face sessions were followed up with six short phone calls interspersed over 3 months. Behavioural change strategies such as goal setting and facilitated problem-solving were used to progress the exercise programme in the frame and optimise adherence. Participants were asked to progress the time they spent standing in the frame so that, by the end of 4 weeks, they were standing for at least 30 min, three times a week. This approach was individualised according to the ability of the person with MS.

### Qualitative study sampling and recruitment
Purposive sampling was used to select participants from those allocated to the intervention group to take part in this embedded qualitative study with the aim of achieving maximum variation. A sampling matrix informed the selection in terms of gender, age, disability level, home environment (eg, from one bedroom flats to houses) and people with and without a standing assistant. Participants were excluded if they did not have the physical capacity to operate an audio recorder or a carer to assist them with this, or had severe communication difficulties preventing them from verbally recording their experiences. From the pilot study by Hendrie et al,[16] it was considered that 20 participants would provide a broad representation of people using the frames under different circumstances so that as much in-depth information as possible about the experience of using a standing frame in the home could be obtained. The decision was made to recruit more people with MS than standing assistants to reflect the possibility that some participants would be living alone, self-managing without an assistant. In total, 12 participants allocated to the standing intervention arm of the RCT, together with 8 standing assistants were invited to participate, and written consent was obtained.

### Data collection
Data were collected using portable, hand-held, audio digital recorders. Participants (people with MS and standing assistants) were provided with, and shown how to operate the recorder and given an opportunity to practise using it. To supplement this, they were provided with illustrated instructions on its use and a written summary to remind them about the purpose of the audio diaries. They were requested to record their experiences of how it felt to stand and use the frame from the first moment they tried it. They were also asked to describe any changes they experienced or witnessed, and include any other comments they wished to make. As our intention was to gather contemporaneous data, participants were asked to record these experiences, if possible, during each stand or as near to the completed standing period as possible. Participants were free to record as many times as they wanted and when they wanted. No further prompts regarding this were given. The audio recorders were collected after the participants had completed the final 36-week RCT assessment. The audio files were downloaded and stored securely. They were transcribed verbatim, dated and anonymised.

### Data analysis
Data were analysed using thematic analysis according to the Braun and Clarke six-phase method of identifying and analysing patterns in qualitative data.[24] In the initial stages, the audio recordings were listened to alongside reading of the transcripts to ensure accuracy. Two members of the research team (WH and RD) read and re-read the transcripts several times and independently assigned relevant initial codes to the data using comment boxes on word versions of the transcripts. To further ensure rigour, a third researcher (JAF) independently listened to, read, re-read and checked the coding of each of the transcripts. In addition, the narrative trajectories were considered over time, exploring whether and how the narratives changed across the trajectory by viewing the diary entries as a whole series rather than solely as fragmented entries.

The next stage of analysis involved reviewing and revising the codes and combining them into themes by looking for meaningful patterns that were relevant to the research aim. This stage was completed longhand rather than using a computer software programme. The assigned codes were considered and critically discussed on several occasions by WH, JAF and RD. Disagreements were resolved through discussion until consensus was obtained. Using this iterative process, themes and subthemes were agreed on, supported by associated key extracts of data that captured the participant voice. Preliminary results were shared with the trial management group, including people with MS, and the extended research team, who were able to reflect and comment on the findings.

The data were analysed as a whole as it was considered that the experiences of the people with MS and their standing assistants were interdependent and entwined.

## Trustworthiness and credibility

The trustworthiness and credibility of the analytical process was optimised through several strategies. The transcripts were independently coded by several members of the research team who were experienced in undertaking qualitative data analysis and detailed discussions were held to ensure decisions could be defended. Triangulation was undertaken with field data gathered from informal, voluntary, exit interviews with all RCT standing group participants who completed the study (61 of the 71 participants allocated to the standing frame group). A summary of the main themes was sent to participants for member checking to ensure credibility of the findings. WH, RD and JAF are all experienced neurological physiotherapists working in the field of MS. In order to minimise the bias this may have brought to the analysis, and to enhance reflexivity, regular trial management meetings were held to enable discussions with the broader research team and MS representatives.

## RESULTS

Twelve people with progressive MS (six female, aged 35–71 years, EDSS 6.5–8.0) and eight standing assistants (four female) kept audio diaries of their experiences of using the Oswestry standing frame. All participants who were invited to participate accepted and none dropped out. Two of the participants encountered technical difficulties using the recorders and, instead, chose to write accounts of their experiences over the duration of the study. Demographic information of participants is presented in table 1.

A total of 155 (range 1–36) diary entries were recorded. More entries were recorded by the individuals with MS (median 8, range 1–36) than the standing assistants (range 1–16 entries). All data were analysed.

Four overarching quoted themes were developed: "Feeling like the old me"; 'Noticing a difference'; "I want to do it right" and "You have a good day, you have a bad day". A number of subthemes were also identified and are presented below, supported by quotes using pseudonyms.

### "Feeling like the old me"

This theme describes how standing reconnected people with their old, more able selves in a positive way, either through changed behaviours brought about by physical improvements or the feelings that were elicited by standing safely upright in the frame. The participants with MS described the enjoyment of standing. As a result, standing made them feel more like the person they used to be. Two subthemes were identified.

#### 'Being upright is really most enjoyable'

Participants talked about the enjoyment they felt standing fully upright again and the feelings that standing evoked. Most commented specifically about the positive impact of supported standing, sometimes from as early as the very first stand.

> Simon has had his standing frame for a week and it has just been the most fantastic thing. He just really enjoys standing up…It seems to have completely changed his life. He is just really excited about life looking forward now, so brilliant…he just loves being upright. *Sophia, standing assistant of Simon EDSS 8.0*

**Table 1** Demographic information of participants (12 people with MS and 8 standing assistants)

| pwMS pseudonym | Gender of pwMS | Age of pwMS | Baseline EDSS of pwMS | Standing assistant pseudonym | Home setting |
|---|---|---|---|---|---|
| James | M | 63 | 6.5 | | Three bed house |
| Justin | M | 68 | 6.5 | | One bed flat |
| Jamie | M | 41 | 6.5 | Claire | Two bed flat |
| Mandy | F | 43 | 6.5 | Keith | Three bed bungalow |
| Sam | F | 69 | 6.5 | Rob | Two bed flat |
| | F | 66 | 6.5 | Thomas | Three bed bungalow |
| Jane | F | 62 | 7.0 | | Three bed house |
| | M | 64 | 7.5 | Liz | Two bed house |
| Joyce | F | 71 | 7.5 | Peter | Three bed house |
| Henry | M | 58 | 8.0 | | Four bed house |
| David | M | 54 | 8.0 | | Three bed house |
| Simon | M | 51 | 8.0 | Sophia | Three bed old cottage |
| | M | 54 | 8.0 | Penny | Four bed house |
| Sarah | F | 53 | 8.0 | | Three bed house |
| Ellen | F | 35 | 8.0 | | Two bed flat |

EDSS, Expanded Disability Status Scale; F, female; M, male; MS, multiple sclerosis; pwMS, person with MS.

Enjoyment from standing was reported throughout the 36 weeks of the trial, even after any potential novelty of using this new piece of equipment had passed. For many, it gave them a feeling of being in control and doing something to help themselves.

It gives me a different kind of freedom because I don't have to cling onto everything … usually I have total lack of confidence, I cling onto everything as I walk, but in the frame it's like a kind of freedom. Although I am strapped in, I am able to move and it's really very enjoyable… it really has given me a sense of liberation. *Jane, EDSS 7.0*

Participants also reported a positive psychological impact of standing even when physical improvements did not appear to have been gained. The pleasure of being upright in standing in itself was a motivator for them to continue.

I don't think the standing frame has helped as far as the MS symptoms are concerned. As far as stretching my muscles, stretching my body and the psychological effect that I am standing which is fantastic. All those side issues are great. *Henry, EDSS 8.0*

### 'A sense of normality'

For some people, the impact of standing upright in the frame gave them a feeling of being 'normal'. People described enjoying the sensation of standing to their full height again and of engaging with previous life roles in standing: a dad listening to his daughter practising her violin, a husband and wife talking in the kitchen.

It gives you a sense of normality…It has been really nice standing in the standing frame looking out of the conservatory watching all the birds on the feeders. *Henry, EDSS 8.0*

And

It's a major, major plus being able to stand up because everything looks the size it always used to and I don't feel like a little tiny seven year old (standing assistant added) [insignificant] boy. *Simon, EDSS 8.0*

On occasions, people provided illustrations of how using the frame had enabled them to achieve something they may not otherwise have managed. For some, this was due to an improvement in symptoms or function (described in the next theme), but for others it was purely standing in the frame itself that facilitated the sense of achievement.

I have been building up for my daughter's wedding, and when it was time for me to make my father-of-the-bride speech they brought the frame in and my carer who was there for the day for me strapped me in, and no problem, I stood for about 13 minutes … read the whole speech, got everyone in tears… *David, EDSS 8.0*

For some, the experience of standing in the frame appeared to give an opportunity to reflect on their past identity,

My friends and I were surfing a lot, that's what we did, we surfed in the summer, surfed in the winter, whenever… the reason I am saying that is that now on the standing frame if I let go of the table or the side arms and lean backwards a little bit I can balance… I can imagine myself standing on the surfboard with my arms down by my side and I just move my body around a little bit in the straps as if I'm moving the board… I am enjoying it tremendously. *Henry, EDSS 8.0*

Family and friends were also affected by seeing their loved ones standing again.

It's really nice to be upright… my mum and her husband came today and she was absolutely amazed. It's a long time since she's seen me standing so we kind of reminisced about the days when I was walking and getting up and about, so that was nice… she had tears in her eyes, bless her…. *Sarah, EDSS 8.0*

### 'Noticing a difference'

This theme captures the variety of positive changes the participants reported in activities such as walking, transfers, posture and sitting balance and in a wide range of MS symptoms including spasms, weakness, muscle stiffness, fatigue and bladder and bowel function. These improvements appeared to increase participants' confidence and enable them to engage more in everyday life. The changes were from across the spectrum of impairments, activities and participations, as illustrated by the following subthemes:

### "My muscles have woken up"

I can truthfully say I felt as if I was using my muscles, the muscles in my calves and thighs were aching but pleasantly as if my body was saying to me 'hello you're using some more muscles that you're not used to using'… consequently I was standing more upright and feeling a little bit more confident about doing things around the house. So, as far as the standing frame is concerned, posture's improved, upper body movement has improved and I'm in a very good frame of mind. *James, EDSS 6.5*

I don't have the spasms I used to have by any means. In fact I have really cut down on the Baclofen, which is the anti-spasm drug. Bowels and things like that… within about a couple of weeks, I suppose, it is so much easier, I can go on demand, so that is really, really good… I've cut down on a load of my medicines. It's the best thing ever. *David, EDSS 8.0*

….we are already noticing a difference… His bowels are fantastic. I don't think he has been constipated in the last few months [since starting the standing programme]. I don't think he wees so much in the

night… He's up for trying new things, going out doesn't seem to be such a problem. *Sophia, standing assistant of Simon EDSS 8.0*

### "We suddenly noticed he was passing the salad bowl"

Participants often talked about functional improvements as they described the day-to-day impact of using the frame.

> I have been able to stand more confidently when I have got up from the toilet and I know that I am able to pull my trousers up… without feeling the need to hold onto anything, so little goals like that I am achieving already. *Jane, EDSS 7.0*

> Just the last couple of days I felt my legs being a little bit stronger and consequently due to that, I've been able to walk a little bit further with less fatigue and it's quite nice feeling that sensation that you know the muscles in your legs are beginning to work. *Justin, EDSS 6.5*

Standing assistants also reported they had observed functional improvements of the person with MS, for example, those associated with increased trunk strength and sitting balance.

> We suddenly noticed he was passing the salad bowl and he reached in with the two salad servers and helped himself. He has never been able to do that before because he has always had to hold on with one hand. And then I caught him piling the plates, reaching across the table, picking up plates, putting them on top of each other and taking them out to the kitchen. *Sophia, standing assistant of Simon EDSS 8.0*

### 'Going out doesn't seem to be such a problem'

Many of the positive changes that people experienced appeared to impact on their confidence to participate with life in a new way which, in turn, gave enjoyment and a sense of achievement.

> I have had no falls since I have been using the standing frame and I have been feeling a little more confident with my balance… Yesterday I went to lunch at a friend's house…I decided to use my husband's arm and a crutch… the improved feeling that I can balance now, it was just really absolutely brilliant… I am so happy that I managed to do it. *Jane EDSS 7.0*

> … I've just been to my [pheasant] shoot today and I'm absolutely amazed, I've been able to stand [perching on the seat of the electric scooter] for a good hour at least, at least 20 minutes at a time, and that's 3 lots for 20 minutes… and I've had a fantastic time. *James, EDSS 6.5*

### "I want to do it right"

Initially, some participants lacked confidence in using the frame and wanted to make sure they were doing it correctly. With increased practise and support from the physiotherapist, however, their confidence grew and they

were able to modify the standing programme to suit their own needs and manage difficulties that arose. This is illustrated by the following subthemes:

### "The physio came round and set me right"

Participants commented that they valued the support and guidance from the physiotherapist in helping them establish a standing programme that worked for them.

> We were… floundering in it… was he standing up completely straight? Was it alright to be leaning back on the back strap? Anyway, so it was very comforting to have the physiotherapist here. *Sophia, standing assistant of Simon, EDSS 8.0*

One person would have preferred increased contact, remarking:

> You do feel left alone a little bit and wondering why you're doing them [the exercises in the standing frame], but I have persevered as far as I can. *Justin, EDSS 6.5*

Interestingly, however, this participant's audio diary entries illustrated how he independently problem-solved issues as they arose and he used the frame regularly over the entire course of the study.

### "I am finding different things as I go along"

People found that they needed to modify their standing routine over time in order to maximise benefit, manage symptoms and gain the greatest enjoyment from it. Making these (often small) changes to the programme appeared instrumental in helping long-term adherence. A number of ideas were described: gradual progression of standing time; standing at different times of the day or on different days of the week; varying the exercises completed or adding functional tasks such as folding the washing.

> I started with a couple of minutes and then worked up to kind of ten minutes, then fifteen minutes, then I was doing my thirty and I am absolutely loving it. *Sarah, EDSS 8.0*

> Mandy decided to use it a little bit later. Normally she uses it mid-afternoon, about half past three, but decided to do it about half past six. She found it a lot easier because that is a better time of day for her. *Keith, standing assistant of Mandy, EDSS 6.5*

### "My back gets a bit achy but it's early days yet"

An important area that some participants talked about was in relation to side-effects that they experienced. These were mainly back and leg aches and pains, which tended to occur early on in the programme, and either resolved completely or reduced in frequency as the individual became more accustomed to standing.

> Today I used the frame for the first time, knees and back a bit sore when standing, but the feeling disappeared when I was back in my chair. *Joyce, EDSS 7.5*

My legs feel quite stiff immediately after I've got out of the frame but that soon passes. *Mandy, EDSS 6.5*

Experience of side effects, even in the early days, was not mentioned by everyone, with a couple of participants specifically noting their absence.

At the moment [week three] there's no side effects for me personally doing them. *Justin, EDSS 6.5*

### "We are definitely not giving the frame back"

At the end of the trial, several of the people with MS reflected on the value they placed on standing in the frame and on how they intended to use it in the long-term. This sentiment was also echoed by many of the standing assistants, and with particular enthusiasm by Sophia.

All in all it has been the most fantastic thing and we are definitely not giving the frame back and he will be using it every day for the rest of his life! *Sophia, standing assistant of Simon, EDSS 8.0*

The acceptability of both the user and standing assistant may be important factors in facilitating long-term use of a frame. In this study, people reported that, after a period of adjustment to this new piece of equipment, they could incorporate this into their weekly routine.

Just to say as a partner [of Simon] and having to help, it is no bother at all. I don't have to haul him up at all. I just wait for him to get in standing position, easily tie him in and sometimes rearrange his feet just to get them exactly right, and then leave him. *Sophia, standing assistant of Simon, EDSS 8.0*

The data revealed that a 'settling in' process was sometimes needed, as people became accustomed to the equipment.

[day one] we had a couple of issues with the standing frame. It is our first time using it on our own.... we used the standing frame again today [day three] we found it much easier to use; we have got the straps set up pretty much where we want them now.... [day seven], so we are getting a lot quicker using it. *Keith, standing assistant of Mandy, EDSS 6.5*

This is an interesting example of the insight provided by the multi-occasion windows that the audio diaries enabled.

Another standing assistant raised two important issues for consideration for anyone contemplating using a frame in their home environment: having sufficient physical ability to move the frame if necessary and having adequate space for it.

The frame itself is quite cumbersome to move so it is better left in situ. In our case it needed to be moved each time to allow a wheelchair or walker access past it. *Thomas, standing assistant of person with EDSS 6.5*

### "You have a good day, you have a bad day"

This theme highlights the challenges faced by people living with a progressive and fluctuating condition when implementing a self-management programme and their expectations of standing. Three subthemes were identified.

#### 'The ups and down of MS'

Many participants talked about how the unpredictability of their condition affected their ongoing ability to engage in the standing programme and that this could change on a day-by-day basis.

I think it is just the nature of my illness, I just, I know only too well that you have a good day, you have a bad day. *Jane, EDSS 7.0*

#### "He really hasn't been feeling well"

This subtheme demonstrates the impact that other illness, infections and environmental conditions can have on someone's symptoms,[4] and in turn, on their ability to consistently engage in a standing programme.

Yesterday for the first time, he couldn't even get up into the frame, which was really scary and thought oh well perhaps it's another bladder infection and we thought we would take another urine sample [to the doctor], and then suddenly yesterday evening he suddenly said 'oh, I am feeling better now'. He did a fifteen minute stand and then got up the bottom step to go to bed… today he seems to be back on track. *Sophia, standing assistant of Simon, EDSS 8.0*

I couldn't use the frame at all last week [week 5], as I had a bad cold and that always leaves me weak and tired as it seems to affect all my muscles. Today I am feeling better, so I used the frame in the afternoon… Last week [week 6] I hurried back into using the frame too quickly after being unwell. I felt comfortable with no pain so I did 35 minutes two days running and was exhausted with back ache. This week I am doing less time, but more often with rest days in between… I feel that the legs and the back are gaining more strength now. *Joyce, EDSS 7.5*

#### "I'm not expecting miracles"

Participants expressed different expectations of the standing frame programme both in terms of their hopes of improvement and the length of time changes may take to happen. In the main, participants appeared satisfied with their experience, although some described an internal dialogue regarding the struggles they had in balancing their aspirations with the reality of their achievements.

I was hoping that by now [six months] I would have noticed something … that would be better, my balance or being able to stand or strengthening my legs or whatever. Maybe it is strengthening my legs, but because they don't work, which is nothing to do with that, it's just the MS damaging the nerves which I

suppose the standing frame isn't going to help is it? I have tried to soldier on, as I usually do… as much as I possibly can… *Henry, EDSS 8.0*

Once again, the multiwindow nature of the audio diary methodology enabled the reader to realise that individuals continued to stand throughout the study timeline, despite the challenges they faced.

### Exit interviews

Qualitative data were also captured in informal exit interviews with the 61 standing group participants who completed the RCT. Interviews were in the region of 10 min duration, and were completed face-to-face at the end of the final study assessment by the research therapist. They were designed to ask briefly about participant experience of the study, aspects that could have been improved, things that they particularly liked and an opportunity to share any other thoughts regarding the study. The main points raised in the exit interviews about the use of the standing frame concurred with the themes and subthemes identified in the audio diary data but without the detail, depth or sense of personal journey. People reported specific physical and psychological changes they had noticed, how they established and modified their standing over time and described issues they faced in terms of both the practicalities of using the frame, and the impact that the variable nature of their symptoms had on maintaining a regular programme.

### DISCUSSION

This embedded qualitative study is, to our knowledge, the first to explore the contemporaneous experience of self-managing a standing frame programme in the home, from the perspective of both the person with progressive MS and their standing assistant. The choice of using audio diaries to facilitate contemporaneous data collection has enabled the reader to gain an insight into both the immediate experience and the standing journey as it unfolded over time. This may help therapists to better understand the experiences of people living with a long-term, progressive and often unpredictable condition when they are asked to carry out a new, self-managed, physical intervention. Other studies have used surveys[10 11 13 14] or interviews[15 16] to explore standing frame use but this methodology has proven helpful in capturing the day-to-day experiences and has provided new detailed insights.

Participants and their standing assistants reported a variety of physical and psychological changes over the 6-month data collection period. Among the range of perceived benefits reported, one very commonly described was that of enjoyment. This was linked by some to feeling a sense of normality and/or freedom, experiences that have also been previously reported in other qualitative studies of supported standing.[13–15] The contemporaneous nature of the data collection revealed these feelings even from early on in the standing programme

and sometimes in the absence of noticing any physical changes. Engagement with an activity is more likely to be sustained over the long-term if that activity is meaningful and enjoyable.[25 26] Our study showed that this was the case for many of the standing participants. Therapists, therefore, have an important role in identifying what the patient considers, for them, is a relevant activity and how they might integrate it into daily life.[27 28]

The subjective reports of improvements in symptoms complements and supports some of the objective results of the RCT, such as those relating to motor function.[9] It is noteworthy, however, that some of the perceived benefits highlighted in this qualitative component, such as improvements in bladder and bowel function and sitting balance were not supported by the objective trial data. There are a number of potential reasons for this: (i) the lack of responsiveness of the standardised measures in detecting small but meaningful improvements for an individual; (ii) the group-based nature of analyses in RCTs, where the focus is on *average* treatment effects[29] and (iii) the restricted range of outcomes that can feasibly be measured within a trial. These issues underline the added value of qualitative work in expanding our understanding of issues which are important to consider when implementing evidence-based interventions into clinical practice.

Gauging the type, timing and level of support that individuals need to sustain effective behaviour change when introducing new equipment is complex and requires careful consideration. For some participants in this study, the range of behaviour change strategies incorporated and level of support provided appeared sufficient to enable them to problem-solve and modify their programme from the outset. These individuals appeared successful in continuing to engage in their standing programme despite the challenging circumstances they faced, which included fluctuating symptoms and adverse events. Others, however, reported they would have valued additional support to gain confidence when learning to use the frame. An example of gaining additional support might be the opportunity to hear the experiences of people in similar circumstances. An output of this qualitative study therefore has been the production of four short films, compiled from these audio diary data, and a narrative account, which can be accessed at www.plymouth.ac.uk/research/sums. In addition, we suggest a number of 'top tips' compiled from the study data which may provide helpful guidance for therapists, people with MS, their family and friends (see box 1).

### Strengths and limitations

Several methodological points have been highlighted by this qualitative study. The choice of using audio diaries enabled data to be collected contemporaneously and longitudinally at multiple windows over a period of time, rather than retrospectively. This approach captured the day-to-day experiences as well as the ongoing challenges faced over time by people when implementing a

**Box 1   Top tips to maximise adherence to a standing frame programme**

► Try to integrate the standing programme into a weekly routine.
► There will be a 'settling in' process which may include short-term aches and pains. This is a normal response to starting a new physical activity and usually improves after a few days.
► Find activities and exercises to do when standing that are enjoyable to undertake.
► There are many ways to modify the programme over time, even when MS symptoms are 'up and down'. Support from a physiotherapist can help with this. Sharing ideas with other people who use a frame can also be useful.
► Not everyone feels up to standing every day—that's ok.
► Frames are about the size of an armchair so space is a consideration. From a practical perspective, it is ideal if they are kept in one place, but they can be moved quite easily with help from an assistant.

self-managed programme while living with a progressive, fluctuating condition. The inclusion of standing assistants gave different perspectives on changes seen as a result of standing, as well as the practicalities of using a frame in the home.

Other strengths of the study are the clear audit trail and variety of strategies to enhance transparency and rigour. Although the findings represent a subsample of participants involved in the RCT, a comprehensive sampling approach was used to ensure representation of viewpoints from a range of participants. In addition, triangulation of the audio data with that obtained from the exit interviews supports the validity of the findings.

It is difficult to know how this audio diary approach fared compared with more traditional interviews. One might surmise, for instance, that the participants may have produced less sanitised accounts of their experiences given the opportunity they were afforded to provide, at their own discretion, a more immediate reaction to a situation in comparison to a more formal interview approach; there is scope for further research regarding this. At a practical level, however, the use of this methodology presented some challenges. Despite showing the person with MS and their standing assistant how to use the audio recorder, and providing written information to support this, some people experienced difficulties with using the audio recorder. Some people with dexterity problems found the small, portable recorder difficult to use and on two occasions, people chose to replace the audio with written notes. On other occasions, it was not until the recorders were returned at the end of the study, that it became apparent that for a few of the participants recordings had only been successfully completed on two or three occasions during the intervention period. Additional systems, such as telephone or email reminders to encourage use of the audio diary, may have helped to minimise this.

While other researchers using audio diary methodology have found ethical challenges in dealing with emotional distress, given the delay in listening to the diaries and thus the inability to offer immediate support in times of distress; we did not experience this when listening to the recordings, although it is an important consideration.[30]

## CONCLUSION

Supported standing in the frame appeared to help people with severe progressive MS experience a sense of normality and enjoyment, which led to them feeling more like their old selves. People reported improvements in physical and psychological symptoms and impairments, which were associated with an increase in activities of daily living and (re-)engagement with activities which were important to them. These positive changes and the enjoyment they derived from standing meant that two-thirds of participants requested to keep the frame at the end of the 36-week trial period in order to continue to use it. Notably, people wanted to continue using the frame even if they had not seen many physical improvements as a result of standing. Physiotherapy support to establish the programme and educate the person with MS and their standing assistant about how to modify it according to their individual and varied needs and symptoms was considered key, as was a recognition of the ups and downs of living with a progressive and fluctuating neurological condition.

**Acknowledgements**  The authors would like to thank all the participants of the SUMS trial; the NHS physiotherapists who implemented the standing programmes and the research assessors (Steve Hooley, Emily Rogers and Danielle Munford).

**Contributors**  JAF, WH, LJ, SC, AB and AH developed the study and contributed to trial design. RD, JAF and WH contributed to data collection, data analysis and writing of the manuscript. LJ, SC and AB contributed to drafts of the paper. All authors approved the final draft of the manuscript.

**Funding**  This research was funded by the National Institute for Health Research (NIHR) (Research for Patient Benefit Programme; PB-PG-1013–32047).

**Disclaimer**  The views expressed in this publication are those of the author(s) and not necessarily those of the NIHR or the UK Department of Health and Social Care.

**Competing interests**  None declared.

**Patient and public involvement**  Patients and/or the public were involved in the design, or conduct, or reporting, or dissemination plans of this research. Refer to the 'Materials and methods' section for further details.

**Patient consent for publication**  Not required.

**Ethics approval**  Ethical approval was gained from the NHS Health Research Authority Committee South West-Frenchay Research Ethics Committee (15/SW/0088). All participants gave written informed consent prior to commencing study activity.

**Provenance and peer review**  Not commissioned; externally peer reviewed.

**Data availability statement**  Data are available on reasonable request. The SUMS study protocol and statistical analysis plan are publicly available at: https://www.plymouth.ac.uk/research/sums. Individual participant data that underlie the results will be made available (after de-identification) on a controlled access basis, subject to suitable data sharing agreements. Requests for data sharing should be made to the Chief Investigator (JAF) in the first instance.

**ORCID iD**
Rachel Dennett http://orcid.org/0000-0003-0400-0502

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
