## [Reviewer comments · BMJ Open]

ARTICLE DETAILS

TITLE (PROVISIONAL)	"I'm in a very good frame of mind": A qualitative exploration of the experience of standing frame use in people with progressive multiple sclerosis
AUTHORS	Dennett, Rachel; Hendrie, Wendy; Jarrett, Louise; Creanor, Siobhan; Barton, Andrew; Hawton, Annie; Freeman, Jennifer

VERSION 1 – REVIEW

REVIEWER	Anna Barabasch Institute for Neuroimmunology and Multiple Sclerosis, University Medical Center Hamburg-Eppendorf, Germany I have received funding from Roche Pharma AG.
REVIEW RETURNED	24-Feb-2020

GENERAL COMMENTS	The authors enrich the results of a RCT of a home-based, self-managed standing frame programme with qualitative data. The manuscript is interesting and it contributes to expanding the knowledge of how people with progressive multiple sclerosis experience the standing frame in their daily life. I also very much appreciate the top tips to maximise adherence to such a standing frame programme. I believe this manuscript should be published by BMJ Open after some points are reviewed and modified. Minor revision is necessary. BMJ Open recommends not to exceed 4000 words. But on the other side, the journal writes that this is flexible. I think it is very difficult to stay with the limited number of words in studies that do qualitative research because the description of the results requires many words. Therefore I would accept the word count of 6192. Please review some points addressed below: Introduction Page 6, L49 "improvement in well-being rose to 88%": What is well-being rose? Do you have made a slip of the pen here? Data analysis I recommend writing "thematic analysis according to Clarke & Braun" because at least many qualitative researchers would then immediately know which method of analysis is meant. Did you use a software program for the organisation and analysis of qualitative data, e.g. MAXQDA? If not, please describe your procedure on how and where you entered/organised and coded your data. Trustworthiness and credibility The authors report that WH, RD and JF are experienced neurological physiotherapists working in the field of MS. Were the
---

	persons involved in data analysis experienced in qualitative research, too? Results Page 17, L15: I recommend to move the quote by Simon "It's a major, major plus being able to stand up because everything looks the size it always used to and I don't feel like a little tiny seven year old (standing assistant added) [insignificant] boy" under the one from Henry (page 16), because it's a good quote for "a sense of normality", but it fits not so much under your description about the reflection of MS patients' on their past identity. Exit interviews I miss some more information about the procedure of the exit interviews. Maybe you could describe your data collection methods as well as the data processing and analysis more in detail. Strengths and limitations Page 31, L. 52: What does "limited data" mean? Was there enough material to answer your research questions?
--	---

REVIEWER	Angeliki Bogosian City, University of London
REVIEW RETURNED	02-Aug-2020

GENERAL COMMENTS	This manuscript describes a very innovative and informative qualitative research. Using audio diaries is a novel approach that can capture participants' changing experiences when using the equipment and given the nature of MS, audio diaries is an appropriate way to do it. The use of qualitative data to create short films is very apt and showcases how research can directly and almost immediately be used to help others in similar circumstances. The manuscript is clearly and comprehensively written. A joy to read! Whereas I don't have any major concerns on the study described, below some points that need more details and clarification: Methods Why did the researchers choose to invite 12 people to keep diaries and eight standing assistants? Is there a justification for the sample size? More clarity is needed around the decision to include standing assistants. It seems that the standing assistants talked a lot about how they thought the pwMS experienced the standing frame. It almost feels their accounts are used alongside pwMS' accounts in order to validate what the pwMS say. Researchers invited 12 people to take part in the diary study, and all of them agreed? If not, more details are needed on sample selection and recruitment. How many people were initially asked to take part, how may decline? Also more details on attrition. Were there any people dropping out, given the effort required to keep a diary? More details are also needed for the diary data collected. How many diary entries were collected (range and mean) overall and
--

	for each participant? At which time points were these diary entries recorded? How long were the diary entries (range and mean) and how long the overall audio file from each participant (range and mean)? It is mentioned in the limitations section of the manuscript that some participants had limited audio data in their diaries, were these participants still included in the analysis? What were the instructions/ diary prompts given to the pwMS? What were the instructions given to the standing assistants? Data analysis: it is not clear how researchers addressed the longitudinal aspect of the diaries; how did they take into account the different time points? Were there any additional techniques or tools they had to use to map data across different time points? Results 'We are definitely not giving the frame back' This is the only theme with only assistants' quotes that is informative and shows the need to include assistants in the study. The theme shows assistants' views on helping the pwMS with the standing frame. Although, it left me assuming that the assistants were the ones that did not want to give the frame back and not the pwMS. Discussion Since researchers used a novel methodology, it would be interesting to further discuss issues around diary-keeping versus more traditional qualitative interviews. It is difficult to know how systematic diary-based information is. Did the authors feel that was the case in this study? Is there a chance that the participants produced 'sanitised' accounts of their experiences and feelings? It's interesting that there were no different themes between the diary entries and the 61 exit interviews, indicating that exit interviews are of equal value to produce similar results and are less cumbersome for the participants and less costly.
--	--

VERSION 1 – AUTHOR RESPONSE

Reviewer comment- Anna Barabasch	Authors response
Page 6, L49 "improvement in well-being rose to 88%": What is well-being rose? Do you have made a slip of the pen here?	We thank the reviewer for highlighting that the word 'rose' may be misinterpreted. We have replaced 'rose' with 'increased' instead (line 129)
I recommend writing "thematic analysis according to Clarke & Braun" because at least many qualitative researchers would then immediately know which method of analysis is meant.	We have added this as suggested for clarity. (line 261)

Data analysis Did you use a software program for the organisation and analysis of qualitative data, e.g. MAXQDA? If not, please describe your procedure on how and where you entered/organised and coded your data.	Thank you for suggesting this addition. We have added further detail on lines 266 and 275 “independently assigned relevant initial codes to the data using comment boxes on word versions of the transcripts.” (line 266) “This stage was completed longhand rather than using a computer software programme.”(line 275)
Trustworthiness and credibility The authors report that WH, RD and JF are experienced neurological physiotherapists working in the field of MS. Were the persons involved in data analysis experienced in qualitative research, too?	Thank you for raising this point. We have added that each of those involved “were experienced in undertaking qualitative data analysis” on line 290.
Results Page 17, L15: I recommend to move the quote by Simon “It’s a major, major plus being able to stand up because everything looks the size it always used to and I don’t feel like a little tiny seven year old (standing assistant added) [insignificant] boy” under the one from Henry (page 16), because it’s a good quote for “a sense of normality”, but it fits not so much under your description about the reflection of MS patients’ on their past identity.	At your suggestion, this quote has been moved to line 376-79.
Exit interviews I miss some more information about the procedure of the exit interviews. Maybe you could describe your data collection methods as well as the data processing and analysis more in detail.	Thank you for this helpful comment. Detail has been added: “Interviews were in the region of 10 minutes duration and were completed face to face at the end of the final study assessment by the research therapist. They were designed to ask briefly about participant experience of the study, aspects that could have been improved, things that they particularly liked and an opportunity to share any other thoughts regarding the study. The main points raised in the exit interviews about the use of the standing frame concurred with the themes and subthemes identified in the audio diary data but without the depth or sense of personal journey”.(line 659-667)
Strengths and limitations Page 31, L. 52: What does "limited data" mean? Was there enough material to answer your research questions?	For clarity we have added detail that “for a few participants recordings had only been successfully completed on two or three occasions during the intervention period.” (line 765)
Reviewer Comments- Angeliki Bogosian	Authors Response

Methods Why did the researchers choose to invite 12 people to keep diaries and eight standing assistants? Is there a justification for the sample size?	Thank you for raising this point for which we have now provided further detail of our justification (lines 233-239). "From Hendrie's pilot study it was considered that twenty participants would provide a broad representation of people using the frames under different circumstances so that as much in-depth information as possible about the experience of using a standing frame in the home could be obtained. The decision was made to recruit more people with MS than standing assistants to reflect the possibility that some participants would be living alone, self-managing without an assistant."
More clarity is needed around the decision to include standing assistants. It seems that the standing assistants talked a lot about how they thought the pwMS experienced the standing frame. It almost feels their accounts are used alongside pwMS' accounts in order to validate what the pwMS say.	Thank you for this point. We have added detail "Exploring the experiences of both people with MS and standing assistants was considered important due to the invaluable role assistants play in the lives of many people with MS. For many people with significant disability, participation in this standing activity would only be possible with help from an assistant. Understanding both perspectives of the frame use therefore was recognised as crucial for successful integration of a standing programme into everyday life." To give more context. (lines 201-7)
Researchers invited 12 people to take part in the diary study, and all of them agreed? If not, more details are needed on sample selection and recruitment. How many people were initially asked to take part, how many decline? Also more details on attrition. Were there any people dropping out, given the effort required to keep a diary?	Thank you for this question. We have added ". All participants who were invited to participate accepted and none dropped out." For clarity (line 304)
More details are also needed for the diary data collected. How many diary entries were collected (range and mean) overall and for each participant? At which time points were these diary entries recorded? How long were the diary entries (range and mean) and how long the overall audio file from each participant (range and mean)?	Thank you for your suggestion of including some further data here. We have added: "A total of 155 (range 1-36) diary entries were recorded. More entries were recorded by the individuals with MS (median 8, range 1 – 36) than the standing assistants (range 1 – 16 entries). All data were analysed." (lines 315-317) Additionally, in response to your questions eight participants (six pwMS and two standing assistants) made entries across the 36 weeks of the study (n=107 entries), seven (four pwMS and three standing assistants) made

	entries up to 26 weeks (n=43) and five (two pwMS and three standing assistants) only made an entry at the end of the study (n=5). We have however we have chosen not included this level of detail within the manuscript.
It is mentioned in the limitations section of the manuscript that some participants had limited audio data in their diaries, were these participants still included in the analysis?	Thank you for this question. For clarity we have added detail that “for a few participants recordings had only been successfully completed on two or three occasions during the intervention period.” (line 765). All data were analysed.
What were the instructions/ diary prompts given to the pwMS?	Thank you for this comment. We have added to the following sections (line 244-55) to add clarity. “Participants (people with MS and standing assistants) were provided with, and shown how to operate the recorder and given an opportunity to practise using it. To supplement this, they were provided with illustrated instructions on its use and a written summary to remind them about the purpose of the audio diaries. They were requested to record their experiences of how it felt to stand and use the frame from the first moment they tried it. They were also asked to describe any changes they experienced or witnessed, and include any other comments they wished to make. As our intention was to gather contemporaneous data, participants were asked to record these experiences, if possible, during each stand or as near to the completed standing period as possible. Participants were free to record as many times as they wanted and when they wanted. No further prompts regarding this were given.
What were the instructions given to the standing assistants?	Thank you for these suggestions. In addition we examined the narrative trajectories over time, exploring whether and how the narratives changed across the trajectory by viewing the diary entries as a whole series rather than as fragmented entries (see for example the quotes for Keith, the standing assistant of Mandy, when describing their ‘settling in’ process under the ‘I want to do it right’ theme (line 579)). We anticipated that the longitudinal nature of the data collection might allow for immediacy of reflections, thus revealing problems and changes as they happened over time but we had no idea at the outset at what time points these would happen. People were asked to
Data analysis: it is not clear how researchers addressed the longitudinal aspect of the diaries; how did they take into account the different time points? Were there any additional techniques or tools they had to use to map data across different time points?	Thank you for these suggestions. In addition we examined the narrative trajectories over time, exploring whether and how the narratives changed across the trajectory by viewing the diary entries as a whole series rather than as fragmented entries (see for example the quotes for Keith, the standing assistant of Mandy, when describing their ‘settling in’ process under the ‘I want to do it right’ theme (line 579)). We anticipated that the longitudinal nature of the data collection might allow for immediacy of reflections, thus revealing problems and changes as they happened over time but we had no idea at the outset at what time points these would happen. People were asked to

	record when they wanted to, rather than at specific points, so it would have been difficult to take set time points at the outset. We have added “In addition we examined the narrative trajectories over time, exploring whether and how the narratives changed across the trajectory by viewing the diary entries as a whole series rather than solely as fragmented entries” (line 268-271) to address the reviewers questions.
Results 'We are definitely not giving the frame back' This is the only theme with only assistants' quotes that is informative and shows the need to include assistants in the study. The theme shows assistants' views on helping the pwMS with the standing frame. Although, it left me assuming that the assistants were the ones that did not want to give the frame back and not the pwMS.	Thank you for this comment. We have added the following to add clarity that this was reported by both people with MS and standing assistants. The quote by Sophia was selected as an example. (lines 556-559) “At the end of the trial, several of the people with MS reflected on the value they placed on standing in the frame and on how they intended to use it in the long-term. This sentiment was also echoed by many of the standing assistants, and with particular enthusiasm by Sophia.”
Discussion Since researchers used a novel methodology, it would be interesting to further discuss issues around diary-keeping versus more traditional qualitative interviews. It is difficult to know how systematic diary-based information is. Did the authors feel that was the case in this study? Is there a chance that the participants produced 'sanitised' accounts of their experiences and feelings?	We agree that this methodology is not commonly used, however it is not within the scope of this article to provide an extended discussion on this, in particular given the word count limitations. Here our aim is to focus on reporting the experiences of the individuals engaged in the standing frame programme, and the impact of this on their daily life. In response the reviewers comments, we have now included within the text (lines 753-759) “It is difficult to know how this audio-diary approach fared compared with more traditional interviews. One might surmise, for instance, that the participants may have produced less sanitised accounts of their experiences given the opportunity they were afforded, at their own discretion, to provide a more immediate reaction to a situation in comparison to a more formal interview approach; there is scope for further research regarding this. At a practical level, however, the use of this methodology presented some challenges”.
It's interesting that there were no different themes between the diary entries and the 61 exit interviews, indicating that exit interviews are of equal value to produce similar results and are less cumbersome for the participants and less costly.	Thank you for this point. Although the broad themes were similar (for example enjoyment, improved function) there was by no means the level of detail or depth that the audio diaries provided. We have added more detail in lines 642-9 “The main points raised in the

	exit interviews concurred with the themes and subthemes identified in the audio diary data but without the depth or sense of personal journey.”
--	---

VERSION 2 – REVIEW

REVIEWER	Angeliki Bogosian City, University of London, UK
REVIEW RETURNED	01-Oct-2020

GENERAL COMMENTS	I have no further comments.
-----------------------------